# Genetic Diversity in the *Suppressyn* Gene Sequence: From Polymorphisms to Loss-of-Function Mutations

**DOI:** 10.3390/biom15071051

**Published:** 2025-07-21

**Authors:** Jun Sugimoto, Danny J. Schust, Takeshi Nagamatsu, Yoshihiro Jinno, Yoshiki Kudo

**Affiliations:** 1Department of Obstetrics and Gynecology, Hiroshima University, Hiroshima 734-8551, Japan; yoshkudo@hiroshima-u.ac.jp; 2Department of Obstetrics and Gynecology, Duke University, Durham, NC 27710, USA; danny.schust@duke.edu; 3Department of Obstetrics and Gynecology, International University of Health and Welfare, Narita Hospital, Chiba 286-8520, Japan; tnag.tky@gmail.com; 4Department of Molecular Biology, University of the Ryukyus, Okinawa 903-2720, Japan

**Keywords:** suppressyn, endogenous retroviruses (ERV), SNP, cell fusion, loss-of-function

## Abstract

The suppressive regulator of cell fusion, *suppressyn*, is specifically expressed in the human placenta and is thought to play a crucial role in trophoblast fusion or syncytialization. Previous studies have suggested that alterations in its expression are associated with aberrant placental development, such as the immature placental morphology observed in Down syndrome, and may contribute to the pathogenesis of fetal growth restriction. While syncytialization in trophoblasts is an essential process for normal placental development, the precise molecular causes of its dysregulation remain poorly understood. In the present study, we aimed to elucidate the potential contribution of genomic variation to the loss of suppressyn function, extending previous analyses of expression abnormalities in perinatal disorders. Through sequence analysis, (1) we identified six polymorphisms within the coding region of the *suppressyn* gene, and (2) discovered that certain deletions and specific amino acid substitutions result in a complete loss of suppressyn-mediated inhibition of cell fusion. Although these mutations have not yet been reported in disease-associated genomic databases, our findings suggest that comprehensive genomic studies of perinatal and other disorders may reveal pathogenic variants of *suppressyn*, thereby uncovering novel genetic contributions to placental dysfunction. It is also anticipated that these findings might direct the development of therapeutic strategies targeting loss-of-function mutations.

## 1. Introduction

The insertion of retroviral sequences into host genomes during evolution has endowed host organisms with new and essential physiological functions [1]. These sequences, known as endogenous retroviruses (ERVs), are a subset of transposable elements, which together account for nearly half of the human genome. Transposable elements, or transposons, are mobile genetic elements broadly categorized into DNA transposons and retrotransposons [2,3]. Retrotransposons, including those derived from human endogenous retroviruses (HERVs), replicate via a “copy-and-paste” mechanism: their RNA intermediates are reverse-transcribed and inserted into new genomic locations. Consequently, HERV sequences now comprise approximately 8% of the human genome [4].

HERVs originated from retroviruses that infected germline cells and became permanently integrated into the host genome. Their basic structure typically includes long terminal repeat (LTR) regions that regulate transcription and coding sequences for viral structural proteins such as *gag*, *pol*, and *env*. Over evolutionary time, HERVs have undergone massive amplification, reaching an estimated 450,000 copies in the human genome [5]. Simultaneously, they have accumulated mutations—including point mutations, deletions, and insertions—that have disrupted many open reading frames (ORFs) and altered their original structure. Despite these disruptions, some of these sequences have been co-opted by the host and are now integral to normal human physiology, resulting in the emergence of functional, virus-derived human proteins. Notable examples include syncytin-1 and syncytin-2, which promote trophoblast cell fusion, and suppressyn, which inhibits cell fusion [6,7,8].

Suppressyn, like the syncytins, is a placenta-specific protein of retroviral origin. It was identified as an endogenous retroviral envelope-derived protein encoded by the *ERVH48-1* locus on human chromosome 21q22.3 [9,10]. This locus includes retroviral coding sequences flanked by LTRs [11] and produces a ~3.3 kb spliced transcript that encodes a 160-amino-acid secreted protein. We previously reported that suppressyn potently inhibits syncytin-1-mediated trophoblast cell fusion and thus designated it “suppressyn” (SUPYN), a fusion-suppressing protein [9]. Suppressyn exerts its function by directly interacting with ASCT2, the receptor for syncytin-1, within the secretory pathway. It is likely that this interaction prevents syncytin-1 binding and results in the formation of a suppressyn-ASCT2 complex that is trafficked to the cell surface, thereby efficiently blocking cell fusion [12,13,14].

Proper regulation of cell fusion is critical for placental development [15,16,17,18,19]. Disruptions in the spatial or temporal regulation of trophoblast fusion can impair the formation of functional placental structures, ultimately leading to compromised fetal growth. Such dysregulation has been implicated in several perinatal complications, including preeclampsia (PE), fetal growth restriction (FGR), and Down syndrome (trisomy 21, TS21).

In TS21 placentas, the villous cytotrophoblast (CTB) layer is immature, with an excessive number of CTB cells compared to gestational age-matched controls [20]. Given that the *suppressyn* gene is located on chromosome 21q22.3, we hypothesized that gene dosage effects in TS21 lead to overexpression of suppressyn, which in turn results in excessive inhibition of trophoblast fusion. Our earlier studies were the first to suggest that genomic alterations in *suppressyn* may contribute to placental dysfunction [21].

In PE placentas, particularly those complicated by FGR, previous reports have demonstrated downregulation of *syncytin-1* and *syncytin-2* expression [22,23,24]. These changes have been linked to epigenetic modifications, such as hypermethylation of the *syncytin-1* promoter [25,26], and genetic variations, including polymorphisms in the 3’ untranslated region (3’UTR) of the *syncytin-2* gene that affect mRNA stability [27].

In the present study, we investigated the potential contribution of *suppressyn* gene variants to disease by examining six polymorphisms identified in public human genome databases for their effects on fusion-suppression activities. Additionally, we analyzed the functional consequences of genomic alterations—such as deletions and point mutations—within the *suppressyn* coding region. Although none of the six identified polymorphisms affected suppressyn’s capacity to suppress syncytin-mediated fusion, we found that specific mutations targeting cysteine residues within the open reading frame led to a complete loss of function. These findings provide the first direct evidence that genetic mutations in *suppressyn* may contribute to disease and underscore the potential value of comprehensive genomic analyses in uncovering novel genetic factors underlying placental disorders.

## 2. Methods

### 2.1. Cell Cultures

HTR-8/SVneo cells (referred to as HTR8 cells throughout for simplicity) [28] were a kind gift from Professor Charles Graham of the Department of Anatomy and Cell Biology at Queen’s University, Kingston, ON, Canada. HTR8 cells were cultured in DMEM (041-29775: Fuji Film, Tokyo, Japan) supplemented with 10% FBS at 37 °C in humidified 5% CO_2_, 20% O_2_. Methods for the establishment of HTR8 cells that stably expressed normal and mutated suppressyn (HTR8-SUPYN) have been described [9], and the cell lines established for this project were cultured in the same medium with the addition of 1 μg/mL puromycin.

### 2.2. Site-Directed Mutagenesis

Primers for deletional mutagenesis are listed in Appendix A. Twenty nanograms of original plasmid (pCAG-Fb1(SUPYN) [9]) were used for amplification with the high-fidelity KOD One polymerase (KMM-101: Toyobo, Osaka, Japan) or PrimeSTAR Mutagenesis Basal Kit (R046A:Takara, Shiga, Japan) as per the manufacturer’s instructions. PCR products were used to directly transform DH5α competent cells, and plasmid DNA was extracted from several clones for sequencing. Plasmids with a 100% sequence match to the specific mutated PCR product were identified and used for transfection.

### 2.3. Stable HTR8 Cell Line Production

Control and site-specific, mutated suppressyn plasmids were transfected to HTR8 cells using Lipofectamine 2000 (11668027: Thermo Fisher Scientific, Waltham, MA, USA) according to the manufacturer’s instruction. After 24 h of incubation, the culture medium was replaced with 3 μg/mL puromycin-containing DMEM complete medium for an additional week in culture. Puromycin-resistant cells were then selected using limiting dilution methods in 96-well plates to obtain single-cell clones stably expressing control and mutated suppressyn.

### 2.4. Immunoprecipitation and Western Immunoblotting

Whole cell extracts from cell lines were prepared in RIPA buffer (50 mM Tris-HCl, pH 8.0, 150 mM Sodium Chloride, 0.5 *w*/*v*% Sodium Deoxycholate, 0.1 *w*/*v*% Sodium Dodecyl Sulfate, 1.0 *w*/*v*% NP-40 substitute with protease inhibitor (165-26021:Fuji film, Osaka, Japan). Extracts were immunoprecipitated with 10 μL of anti-Flag M2 agarose (A-2220; Sigma-Aldrich, St. Louis, MO, USA) according to the manufacturer’s recommendations. Captured proteins were separated by standard SDS–PAGE and analyzed by immunoblotting with the following specific primary antibodies: monoclonal anti-Flag M2 antibody (F1804; Sigma-Aldrich, St. Louis, MO, USA), polyclonal anti-DDDDK-tag (PM020; MBL, Tokyo, Japan), polyclonal anti-ASCT2 antibody (8057: Cell signaling technology, Danvers, MA, USA), and monoclonal anti-β-actin (A5441: Sigma-Aldrich, St. Louis, MO, USA).

### 2.5. Transient Transfections of Syncytin-1

1.5 × 10^5^ cells from HTR8 cell lines (HTR8, HTR8-vec, HTR8-SUPYN wild-type, or with mutagenesis) were used to seed 24-well plates. After 24 h of incubation, cells were exposed to 25–125 ng of a plasmid vector driving expression of syncytin-1 for transient transfection using Lipofectamine 2000 (11668027: Thermo Fisher Scientific, Waltham, MA, USA) and incubated for 24 h. Cells were trypsinized and collected as single-cell suspensions in PBS.

### 2.6. Detection of Cell Fusion Rate by Flow Cytometry

Cells were analyzed by flow cytometry (FACSVerse; BD Biosciences, Franklin Lakes NJ, USA) using forward and side scatter (FSC vs. SSC) parameters to separate populations based on cell size and internal complexity as described previously in detail [9]. A reference population representing standard cell size was defined using non-transfected HTR8 cells. Based on this reference, a gate was set to identify cells with larger size, corresponding to fused cells (referred to as the “fused” gate: Appendix A). HTR8 cells transfected with an empty vector, as well as those transfected with increasing amounts of syncytin-1 plasmid (25–125 ng), were analyzed in the same manner. The percentage of fused cells was quantified as the proportion of cells assigned to the fused gate. Similarly, HTR8-vec, HTR8-SUPYN wild-type, and HTR8-SUPYN mutant cells were analyzed, and the percentage of fused cells was determined by applying the same fused gate as defined for the HTR8 controls. Fusion indices were calculated by normalizing the percentage of fused cells in each condition to that of the corresponding untransfected (un-T.F.) cells, which was set to 1.

### 2.7. Trans-Well Analysis for Secreted Suppressyn Protein

A total of 5 × 10^5^ cells from HTR8 cell lines (HTR8, HTR8-vec, HTR8-SUPYN wild-type or c.427-1 and c.427-2) were seeded on separate 0.4 μm 6-well transwell plates (upper chamber) with HTR8 cells seeded in each of the lower chambers (657640: Greiner Bio-one, Kremsmunster, Austria). After 48 h of incubation with secreted suppressyn exposure across the trans-well insert, the lower HTR8 cells were treated with 750 ng of a plasmid vector driving expression of syncytin-1 for transient transfection using Lipofectamine 2000 (11668027: Thermo Fisher Scientific, Waltham, MA, USA) and incubated for 6 h. After an additional 24 h incubation with the cells in the upper chamber, the lower HTR8 cells were trypsinized and collected for flow cytometry analysis.

### 2.8. Statistical Analyses

Data are expressed and plotted as means +/− standard deviations. Means were compared using Mann–Whitney U-tests. Statistical significance was defined as * *p* < 0.05 or ** *p* < 0.01, as indicated.

## 3. Results

### 3.1. Analysis of Suppressyn Polymorphisms Using the 1000 Genomes Database

To investigate the presence of known genetic variants within the *suppressyn* coding sequence, we queried the 1000 Genomes Project database [29] for single nucleotide variants (SNVs) located within the region encoding the 160-amino acid suppressyn protein. This analysis identified six nucleotide substitutions (SUPYN IDs: c.47, c.79, c.385, c.394, c.427, c.449) (Figure 1a, Appendix A). Among these, three variants—c.79, c.394, and c.449—were classified as common polymorphisms with a minor allele frequency (MAF) >1%, whereas c.47, c.385, and c.427 were rare variants. All six substitutions were missense mutations, potentially altering the amino acid sequence and suggesting possible functional effects on fusion suppression.

To assess the functional impact of these variants, we generated stable HTR8 cell lines expressing suppressyn constructs harboring each of the six individual nucleotide substitutions (Figure 1a; see also Sugimoto et al., Sci Rep, 2013 [9]). Protein expression was confirmed by immunoblotting, revealing suppressyn bands of the expected molecular weight in both cell lysates and culture supernatants for most constructs (Figure 1b,c). However, in the c.427 clone, the secreted suppressyn protein appeared smaller than the wild-type, indicating a possible defect in post-translational modification.

We next evaluated fusion-suppression activities using a FACS-based assay system, which quantifies cell fusion following dose-dependent transfection of syncytin-1 expression constructs. As shown in Figure 1d,e, all suppressyn variants retained their ability to inhibit syncytin-1-mediated cell fusion at levels comparable to the wild-type. Notably, minor but statistically significant increases in fusion were observed in both independent single clones c.79, c.394, and c.385 at high syncytin-1 expression levels, indicating marginally reduced fusion-suppression activity.

Focusing on the c.427 variant, which exhibited aberrant molecular size in its secreted form, we further assessed its fusion-suppression capacity. Immunoblot analysis demonstrated that the secreted c.427 suppressyn protein retained the same molecular weight as its intracellular counterpart (Figure 2a). This size shift is likely due to impaired O-glycosylation at threonine 143 (T143), a predicted O-glycosylation site (Appendix A). To confirm that the amino acid substitution of threonine to alanine affects O-glycosylation, we also introduced two additional mutations: a threonine-to-serine substitution that was expected to cause partial disruption of O-glycosylation and a threonine-to-valine substitution that was anticipated to result in a complete loss of O-glycosylation. Analysis of these mutant constructs revealed glycosylation patterns consistent with our predictions, thereby identifying T143 as a critical residue for O-glycosylation (Appendix A). To determine whether this modification defect affects function, we performed a transwell-based assay using 0.4 μm inserts to measure fusion inhibition by secreted suppressyn (Figure 2b). The fusion-suppression activity of the c.427 variant was indistinguishable from wild-type suppressyn, which exhibited an approximately 30% reduction in cell-cell fusion (Figure 2c).

Finally, since variants c.79 and c.449 have been reported to co-occur on the same allele, we assessed the functional impact of a dual-substitution suppressyn construct. Protein expression levels were unaffected for both cell-associated and secreted forms, and the fusion-suppression capacity remained comparable to that of the wild-type protein (Appendix A).

### 3.2. Identification of Loss-of-Function Mutations and Deletions in Suppressyn

Given that none of the SNV naturally occurring substitutions identified in the 1000 Genomes Project database affected suppressyn function, we next aimed to identify mutations that could lead to loss of function. Previous analyses revealed that the 160-amino acid suppressyn protein contains several critical functional domains, including a signal peptide sequence required for secretion (Appendix A) and a syncytin-1-like motif predicted to mediate interaction of suppressyn protein with ASCT2 (Figure 3a or Figure 4a; see also Sugimoto et al. SciRep, 2013 [9]). To systematically assess the functional relevance of these domains, we divided the suppressyn protein into six internal fragments (F2–F6) and generated seven deletion constructs, each containing one or two of these fragments. All constructs also include the N-terminal six amino acids of suppressyn, including the initial methionine, which are designated as fragment F1. Fragments F2 through F6 correspond to amino acid residues 7–39, 40–66, 67–117, 118–144, and 145–160, respectively. A Flag epitope was added to the C-terminus of each construct to allow detection using western immunoblots. The seven deletion constructs were used to generate stable HTR8 cell lines expressing each variant (HTR8-SUPYN Del #1–Del #7).

As shown in Figure 3b, western blot analysis revealed that the molecular weights of intracellular suppressyn proteins corresponded to the predicted sizes based on their respective deletions. Notably, uncleaved pre-proteins—prior to signal peptide cleavage—were only detectable in constructs containing the F2 fragment (Figure 3b, asterisk). Similarly, secreted forms of suppressyn were only observed in constructs containing F2 (HTR8-SUPYN wild-type, Del #2, #3, #4, and #7), suggesting that the F2 region is essential for proper secretion.

To examine the interaction between suppressyn and its receptor ASCT2, we performed co-immunoprecipitation assays. While wild-type suppressyn co-precipitated with ASCT2, none of the deletion variants exhibited detectable association (Figure 3b, lower panel), indicating that these deletions disrupt receptor binding. Consistent with this, all deletion constructs failed to inhibit syncytin-1-mediated cell fusion in the FACS-based assay, supporting the essential nature of intracellular ASCT2 interaction for proper suppressyn-mediated inhibition of cell fusion (Figure 3c).

Finally, we investigated the contribution of specific amino acid residues to suppressyn function by focusing on cysteine residues, which are hypothesized to mediate multimerization through disulfide bond formation. Suppressyn contains eight cysteines, although the first resides within the signal peptide and was therefore excluded from analysis (Figure 4a). We individually substituted the remaining seven cysteines with alternative amino acids and established stable HTR8 cell lines expressing each mutant (HTR8-SUPYN Cys#1–#7).

As shown in Figure 4b, all cysteine mutants expressed suppressyn proteins of the expected size in both cell lysates and culture supernatants, although expression levels varied. Functional assays revealed that all mutants, except Cys#5, lost their ability to suppress syncytin-1-mediated cell fusion (Figure 4c). These functional results correlated with loss of ASCT2 interaction, as assessed by co-immunoprecipitation (Figure 4b). Thus, cysteine residues play a critical role in maintaining suppressyn’s functional conformation and receptor-binding ability, with Cys#5 being the only non-essential residue among those tested.

## 4. Discussion

Suppressyn is a placenta-specific protein that acts as a negative regulator of cell–cell fusion and is believed to play a critical role in placental morphogenesis. As such, its dysregulation may have significant implications for pregnancy establishment and maintenance. Previous studies have primarily focused on expression-level changes in suppressyn; however, the present study aimed to explore the relationship between genomic variation within the *suppressyn* gene and disease pathogenesis.

Using the 1000 Genomes Project database, we identified six single-nucleotide variants (SNVs) within the coding region of the *suppressyn* gene. Three of these (c.79, c.394, and c.449) appeared to be common polymorphisms, whereas the remaining three (c.47, c.385, and c.427) were rare and potentially pathogenic. Among these, the c.427 variant altered the size of the secreted suppressyn protein, although it did not significantly impact fusion suppression activity or interaction with ASCT2. Given that c.427 affects a predicted O-glycosylation site, the observed secretion abnormalities raise the possibility of additional functional consequences, warranting further investigation.

Compared to the remarkable functional resilience of naturally occurring SNVs of *suppressyn*, our site-directed mutagenesis studies revealed the structural and functional fragility of the protein itself. Deletions involving the putative ASCT2-interacting motif invariably resulted in a complete loss of fusion suppression activity. Even the isolated deletion of the N-terminal signal peptide led to functional inactivation, as evidenced by the absence of the protein in the culture supernatant and the loss of ASCT2 binding. These results suggest that suppressyn must first enter the secretory pathway, where it likely interacts with ASCT2 prior to glycosylation—consistent with previous reports on suppressyn’s mechanism of action [12]. Supporting this notion, a unique precursor band corresponding to the uncleaved signal peptide was only observed in constructs containing the F2 fragment (Figure 3b, asterisk), confirming that suppressyn undergoes cleavage of a ~39-amino-acid signal peptide.

Taken together, these results indicate that deletion of any major domain within the 160-amino-acid suppressyn protein results in a complete loss of function. Further, the small number of identified, naturally occurring SNVs, combined with their apparent lack of functional consequence, argue for the great importance of *suppressyn* sequence conservation over time.

We also demonstrated the critical role of cysteine residues in suppressyn function, particularly with respect to disulfide bond formation and potential multimerization. Suppressyn harbors seven key cysteines (excluding one located within the signal peptide). Site-specific mutagenesis of these residues revealed that six out of seven are essential for ASCT2 interaction and fusion suppression. Only the Cys#5 mutant (residue 95) retained functional activity and receptor binding. These findings strongly suggest that disulfide bonds formed by these six cysteine residues are indispensable for the proper structural conformation required for ASCT2 interaction. Given the established importance of cysteine residues in viral envelope protein function, including in retroviral infection [30], our data reinforce the evolutionary conservation and physiological relevance of these motifs in endogenous retroviral proteins that have been repurposed for host function.

To assess the clinical relevance of these findings, we screened public genomic repositories, including the Beacon Network (https://beacon-network.org/#/ accessed on 12 November 2024), for disease-associated mutations within *suppressyn*. Although no cysteine-altering mutations have yet been reported, future whole-genome sequencing efforts across diverse disease cohorts may reveal such variants. Notably, a missense SNV corresponding to SUPYN-Cys#2 (rs2058796953) is listed in the NCBI dbSNP database (Appendix A), suggesting that a subset of humans may harbor functionally compromised *suppressyn* alleles. Such mutations may not only contribute to perinatal complications but could also underlie early pregnancy loss or implantation failure, which can be particularly hard to detect clinically. Therefore, comprehensive genomic analyses focusing on severe reproductive disorders are warranted.

## 5. Conclusions

While a direct link between suppressyn mutations and specific pathologies remains to be established, our findings define key structural determinants required for suppressyn function. In particular, the signal peptide sequence and conserved cysteine residues are critical for its proper processing and biological activity—with a notable role in the inhibition of cell fusion. These insights provide a molecular framework for understanding suppressyn’s role in human reproduction and highlight its potential as a target for therapeutic intervention in pregnancy-related disorders.

## Figures and Tables

**Figure 1 biomolecules-15-01051-f001:**
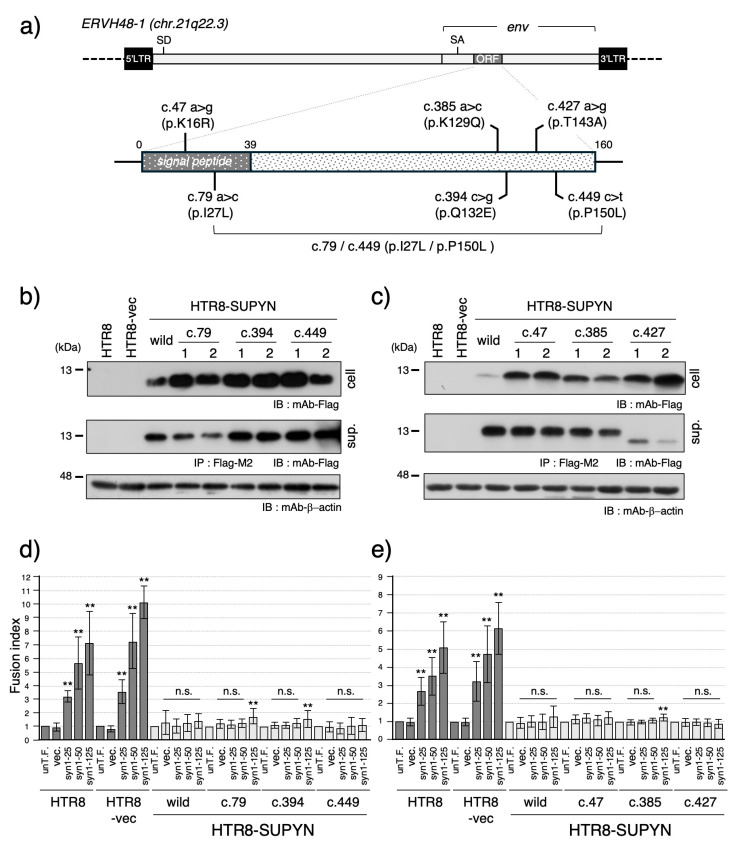
Functional analysis of suppressyn single nucleotide variants (SNVs) identified in the 1000 Genomes Project database. (**a**) Schematic representation of six *suppressyn* SNVs (c.47, c.79, c.385, c.394, c.427, and c.449) mapped within the coding region, as identified in the 1000 Genomes Project database. The single nucleotide substitutions at c.79 and c.449 have been reported to co-localize on the same allele and are therefore connected by a line and labeled as c.79/c.449 (p.I27L/p.P150L) in this scheme. (**b**) Western blot analysis of suppressyn protein expression in HTR8 cells stably expressing suppressyn variants containing SNVs (c.79, c.394, and c.449). Cell lysates and conditioned media (after immunoprecipitation with anti-Flag M2 agarose) were analyzed to detect the expected protein bands. The numbers 1 and 2 shown in the figure indicate technical duplicates generated using two independent clones. ‘cell’ indicates the cell lysate, and ‘sup.’ indicates the culture supernatant. (**c**) Western blot analysis of suppressyn protein expression in HTR8 cells stably expressing suppressyn variants containing SNVs (c.47, c.385, and c.427), as performed in (**b**). (**d**) Cell fusion assays were performed using suppressyn variant-expressing cells as in (**b**), transfected with increasing amounts (25, 50, or 125 ng) of syncytin-1 expression vector. Fusion rates were quantified by FACS and normalized to the untransfected control (set as 1.0). The abbreviation ‘un-T.F.’ refers to untransfected cells. ‘cell’ indicates the cell lysate, and ‘sup.’ indicates the culture supernatant. Error bars represent the standard deviation (SD). (**e**) Cell fusion assays were performed using suppressyn variant-expressing cells from (**c**), analyzed as in (**d**). Statistical significance compared to the corresponding vector-only control was assessed using the Mann–Whitney U test. *p* < 0.01 by **, and “n.s.” denotes not significant. All experiments were performed in biological duplicates and repeated independently at least three times. Original full-length blots for SDS-PAGE analyses are provided in Appendix A.

**Figure 2 biomolecules-15-01051-f002:**
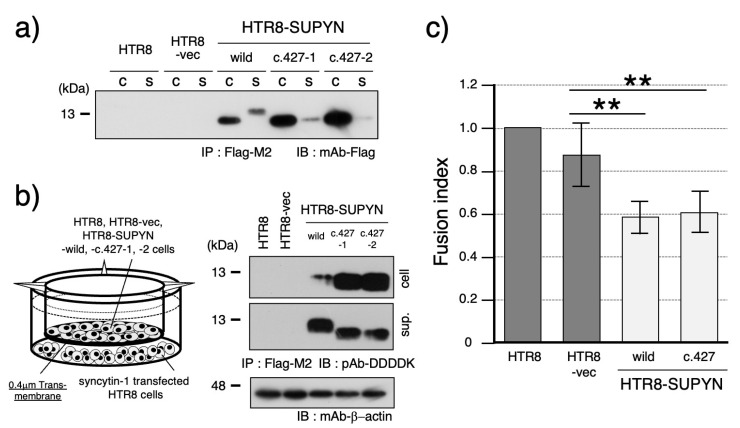
Functional characterization of the aberrantly secreted suppressyn protein harboring SNV c.427. (**a**) Western blot analysis comparing the molecular weight of intracellular and secreted suppressyn proteins containing the SNV c.427. Samples from both cell lysates and conditioned media were immunoprecipitated using anti-Flag M2 agarose prior to detection. ‘C’ indicates the cell lysate, and ‘S’ indicates the culture supernatant. (**b**) Schematic of the transwell-based cell fusion inhibition assay used to evaluate the function of secreted suppressyn containing the c.427 SNV. Suppressyn-expressing cells were cultured in the upper chamber of a 0.4 μm transwell insert, while HTR8 cells transfected with syncytin-1 were placed in the lower chamber. Representative western blot analysis of suppressyn protein expression in both cell lysates and conditioned media, following immunoprecipitation with anti-Flag M2 agarose. ‘cell’ indicates the cell lysate, and ‘sup.’ indicates the culture supernatant. (**c**) Quantification of cell fusion in the syncytin-1-transfected HTR8 cells cultured in the lower chamber of the transwell plate. Data are presented as mean fusion rates, and statistical significance was assessed using the Mann–Whitney U test. *p* < 0.01 is indicated by **. All assays were performed in biological duplicates and independently repeated at least three times. Error bars represent the standard deviation (SD). The numbers 1 and 2 (c.427-1, c.427-2) shown in the figure indicate technical duplicates generated using two independent clones. Original full-length SDS-PAGE blots are provided in Appendix A.

**Figure 3 biomolecules-15-01051-f003:**
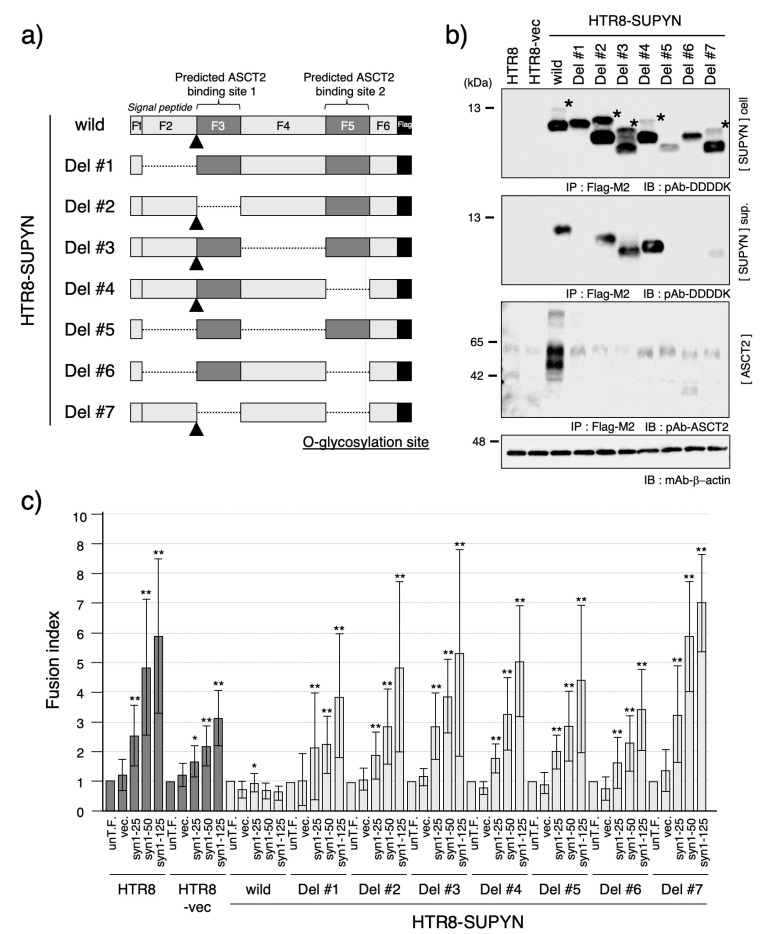
Functional analysis of suppressyn deletion constructs. (**a**) Schematic representation of the suppressyn protein deletion constructs. The full-length suppressyn sequence (160 amino acids) was divided into six fragments (F1–F6) based on predicted structural and functional domains. F1 and F2 correspond to the 39-amino acid signal peptide; F3 and F5 encompass putative ASCT2-binding motifs; and F5 also includes a predicted O-glycosylation site. The signal peptide cleavage site is indicated by a triangle. Deleted regions in each construct are indicated by dotted lines. (**b**) Western blot analysis of suppressyn protein expression in HTR8 cells stably expressing each deletion construct. Immunoprecipitation was performed using anti-Flag M2 agarose prior to detection. Upper panel: intracellular protein; middle panel: secreted protein in conditioned media; lower panel: ASCT2 detection by immunoblotting. Constructs containing F2 fragments retained signal peptide cleavage activity, indicated by asterisks. ‘cell’ indicates the cell lysate, and ‘sup.’ indicates the culture supernatant. (**c**) Cell fusion assay results using syncytin-1 transiently-transfected HTR8 cells stably expressing each suppressyn deletion mutant. Fusion inhibition activity was evaluated by FACS analysis, and statistical significance was assessed using the Mann–Whitney U test. * *p* < 0.05; ** *p* < 0.01. All experiments were performed in duplicate and independently repeated at least three times. The abbreviation ‘un-T.F.’ refers to untransfected cells. Error bars represent the standard deviation (SD). Full-length SDS-PAGE images for all immunoblots are shown in Appendix A.

**Figure 4 biomolecules-15-01051-f004:**
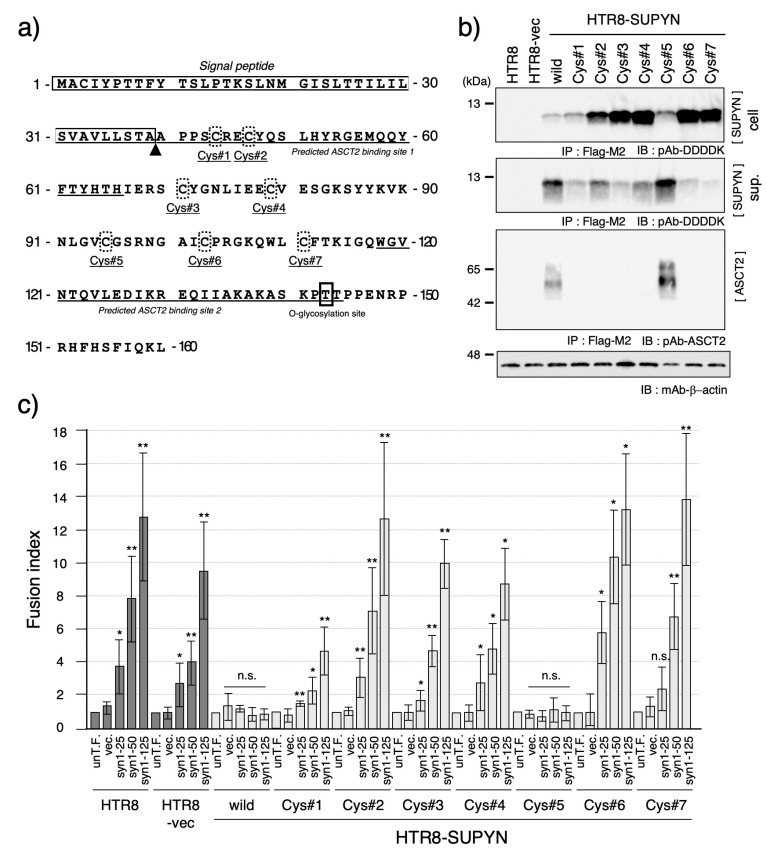
Functional analysis of cysteine residue mutants of suppressyn. (**a**) Schematic representation of the amino acid sequence of suppressyn, highlighting the positions of the seven cysteine residues (Cys#1–#7, shown in dashed boxes). The boxed region indicates the signal peptide, and the triangle denotes the predicted cleavage site. Underlined sequences represent the putative ASCT2-binding regions. (**b**) Western blot analysis of suppressyn proteins carrying point mutations at individual cysteine residues expressed in HTR8 cells. Proteins were detected after immunoprecipitation using anti-Flag M2 agarose. Upper panel: intracellular protein; middle panel: secreted protein in conditioned media; lower panel: ASCT2 detection by immunoblotting. ‘cell’ indicates the cell lysate, and ‘sup.’ indicates the culture supernatant. (**c**) Cell fusion inhibition assay using syncytin-1 transiently transfected HTR8 cells stably expressing each cysteine mutant-containing suppressyn protein. Fusion suppression activity was quantified by FACS analysis. Statistical significance was assessed using the Mann–Whitney U test. * *p* < 0.05; ** *p* < 0.01; n.s., not significant. All experiments were performed in duplicate and repeated independently at least three times. The abbreviation ‘un-T.F.’ refers to untransfected cells. Error bars represent the standard deviation (SD). Full-length SDS-PAGE images of the immunoblots are provided in Appendix A.

## Data Availability

The original contributions presented in this study are included in the article. Further inquiries can be directed at the corresponding author.

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
