# Peer review of "Genetic Diversity in the Suppressyn Gene Sequence: From Polymorphisms to Loss-of-Function Mutations"

_biomolecules, 2025, doi:10.3390/biom15071051_

Round 1
Reviewer 1 Report
Comments and Suggestions for Authors
This study is a follow-up investigation by Sugimoto et al.'s laboratory. They previously identified Suppressyn (SUPYN), a protein derived from an endogenous retrovirus (HERV), as being specifically expressed in the human placenta. SUPYN is a secreted protein consisting of 160 amino acids and has previously been shown to play a crucial role in trophoblast syncytialisation. The authors had previously reported that SUPYN effectively inhibits syncytin-1-mediated trophoblast cell fusion.
In this study, the authors examine the functional consequences of mutations in the SUPYN gene, including amino acid substitutions and deletions. Specifically, they created polymorphic variants (identified in the 1000 Genomes Project database) and assessed the effect of these variants on fusion suppression using a cell fusion assay.
Six polymorphisms identified from public human genome databases were initially tested. Although all six were missense mutations, which are typically assumed to influence protein function, the authors found that none of these substitutions altered SUPYN's capacity to suppress syncytin-mediated cell fusion. A cell fusion assay was utilised throughout the study for functional characterisation.
The authors also examined the key functional domains of SUPYN. These were: (i) the N-terminal signal peptide required for secretion; (ii) a syncytin-1-like motif, which is believed to facilitate interaction with the ASCT2 receptor; (iii) the mutagenesis of cysteine residues within the open reading frame. These targeted mutations resulted in a complete loss of function.
This study clearly presents valuable functional insights into the structural features of SUPYN.
Major concerns:
- The study has the limitation that its conclusions are based exclusively on a single functional assay (cell fusion), which restricts the interpretation of certain findings. Conclusions would be strengthened by additional biochemical and cellular assays.
- The c.79 variant appears to be expressed at lower levels than the wild type and other mutants. However, the manuscript lacks quantitative expression data, such as densitometry of Western blots or mRNA quantification. The authors should quantify expression, and clarify whether the lower expression is due to reduced transcription, mRNA instability, impaired translation or increased protein degradation. As the overexpression of SUPYN has been found to have an inhibitory effect, the question arises as to what the consequences of a lower dosage could be.
- The observed size shift for the c.427 variant is attributed to impaired O-glycosylation at the predicted glycosylation site of threonine 427. While this prediction is plausible, it requires experimental validation. The following T427 substitutions are recommended: (i) T→S(erine), which is expected to partially preserve glycosylation, and (ii) T→A(lanine) or T→V(aline), which are expected to abolish glycosylation. Differential size shifts across these substitutions would lend weight to the hypothesis.
Minor comment
"It is believed that HERVs originated from retroviruses that infected germline cells and became fixed in the genome." It is not simply believed… For accuracy, this should be revised to read: 'HERVs originated from retroviruses that infected germline cells and became permanently integrated into the host genome.'
Reviewer 2 Report
Comments and Suggestions for Authors
Suppressyn (SUPYN) is a placental protein that originates from the env region of HERV-Fb1, a member of the human endogenous retrovirus (HERV) family. It functions as a negative regulator of trophoblast syncytialization. In this study, Sugimoto et al. examined how genomic variation may contribute to the loss of suppressyn function. To investigate this, they generated stable HTR-8/SVneo trophoblast cell clones that express mutant forms of SUPYN. They analyzed six single nucleotide variants (SNVs), deletion mutants and cysteine residue variants to assess their impact on SUPYN’s ability to inhibit syncytin-mediated cell fusion. Cell fusion assays were performed transiently transfecting suppressyn mutant expressing clones with increased amount of syncytin-1 expression constructs. The results showed that none of the six SNVs (missense mutations) significantly affected SUPYN’s fusion-inhibitory activity. All deletion mutants were unable to suppress syncytin-1-mediated fusion and failed to bind the ASCT2 receptor. This highlights the critical role of these regions in SUPYN’s function. Notably, F2 region was reported to be essential for the secreted form of the protein. All cysteine mutants, except Cys#5, lost their ability to inhibit cell fusion and to interact with ASCT2. Overall, the manuscript provides novel insights into how certain types of mutations can adversely affect the function of the suppressyn protein.
Comments and Suggestions for the authors:
Page 2. “In our previous work, we also identified a marked decrease in suppressyn expression in PE placentas”. Add reference.
Page 4: “HTR8-SUPYN wild or c423-1, c423-2” should be “HTR8-SUPYN wild type or c.427-1, c427-2”
In supplementary Figure S1: The line showing “amino acid substitution” should be listed in reverse order.
Figure 1a: What does the line shown as c79/c449 (pI27L/p.P150L) highlight? It is not clear to me.
Figure 1b and c: What do “1” and “2” mean in the lines? Are there two clones for each variant? Please specify.
Figure 1d and e: Captions to graph require bigger fonts. What does it mean unT.F.?
Legend Figure 1e: Cell fusion assays were performed using…
Page 6: “minor but statistically significant increases in fusion were observed in clones c.79, c.394..”. However, observing the figure, it is only true for the clone c.79-1 and not for the c79-2. This could be due to experimental variability rather than the specific mutation. The same applies to variant c.385 and to some extent also to variant c.394. Once again, do numbers 1 and 2 refer to clones or biological duplicates? If they are duplicates, they should be grouped together.
The positions of amino acid deletions should be reported in the manuscript, for example as “from – to”.
Page 6: O-glycosylation at threonine 427 (T427), should be reported as threonine 143 (T143).
Some typos:
Page 4: SUPN IDs should be SUPYN IDs.
Supplementary Figure S3 in the graph, c.79/499 should be c.79/449.
Reviewer 3 Report
Comments and Suggestions for Authors
This is an overall interesting paper. The manuscript is well written and the results have been presented clearly. However, the following points should be amended:
- Page 2, last paragraph: The authors cite reference 10 for the flow cytometry. However, even in this reference the exact description of the flow cytometry protocol and calculation of the fusion index is missing. Therefore, the authors should include (i) more details about the flow cytometry, (ii) representative FSC-SSC graphs of cells with high and low fusion indices, and (iii) the exact procedure for calculation of the indices.
- Page 3, line 8: The authors mention that methods for stable expression of SUPYN in HTR8 cells have been described. This statements need an appropriate reference.
- Fig 1: the x axis labels in d and e are two small. In the printed version, they are not readable. Please increase font size.
- Fig 1: in b and c, the abbreviation “sup.” should be mentioned in the legend.
- Fig 1: what is the meaning of “1” and “2”. Are these the mentioned duplicates?
- Page 6, line 14 ff: “Notably, minor but statistically significant….”. It seems that this statement is true only for c.394 because in the case of c.385 and c.79, significance is only observed in one of the duplicates.
- 2 c: why are the fusion indices all <1? According to fig 1 the indices for the grey bars should be higher. It seems that in this experiment the control without syncytin-1 is missing. The authors should include this control and calculate the indices as in fig 1.
- Page 7: “Finally…”. The co-occurrence of the polymorphisms is depicted in figure 1a but is not explained in the legend. Please include the explanation of the bracket in the legend to fig 1 a.
- Page 7, last paragraph: The sentence starting with “To systematically assess…” contains a typo (“all of which all of which”).
- Fig 3 c: again, the x axis labels are two small
- Fig 4 c: again, the x axis labels are two small.
- In all figures the meaning of the error bars (standard errors, standard deviations?) should be explained in the legends.
- What is the basal expression of syncytin 1 (and probably other fusogenic ERV) in HTR8 cells?
- Please use the hyphenation “poly-morphisms” instead of “pol-ymorphisms” (title and abstract).
Round 2
Reviewer 1 Report
Comments and Suggestions for Authors
I do not have more comments.
Reviewer 3 Report
Comments and Suggestions for Authors
The authors have addressed all points from the first review. I have no further suggestions.